# Factors associated with wasting among pediatric cancer patients aged 2–17 years at Uganda cancer institute: A cross-sectional study

Daisy Wannyana[1,2]*, Arthur Bagonza[1], Sandrah Joyce Mwima[1,2], Christine Nalwadda[1], Rawlance Ndejjo[3,4]

1 Department of Community Health and Behavioural Sciences, School of Public Health, Makerere University Kampala, Uganda, 2 Department of Public Health and Nutrition, Faculty of Health Sciences, Victoria University, Kampala, Uganda, 3 Department of Disease Control and Environmental Health, School of Public Health, Makerere University, Kampala, Uganda, 4 Department of Preventive Medicine, College of Medicine, Korea University, Seoul, South Korea

* wannidaisy@gmail.com

## Abstract

### Introduction

Wasting is a major concern among pediatric cancer patients and significantly affects treatment outcomes and quality of life. However, limited data exist on the prevalence of wasting and its associated factors in low-income contexts. This study determined the prevalence of wasting and its associated factors among pediatric cancer patients aged 2--17 years at the Uganda Cancer Institute.

### Methods

An institutionally based, cross-sectional study was conducted among 270 systematically randomly selected caregiver–child pairs. Univariate, bivariate, and multivariable analyses were conducted using STATA version 14. Variables with p-value < 0.05 were considered statistically significant.

### Results

Among 270 pediatric cancer patients aged 2–17 years, 27.4% (n = 74) were wasted. Children aged 5 years and older had a 20% higher prevalence of wasting (aPR = 1.2; p = 0.002). Cancers near the gastrointestinal tract were associated with a 10% greater prevalence of wasting (aPR = 1.1; p = 0.028). Wasting was lower by 20% among children whose caregivers had tertiary education (aPR = 0.8; p = 0.002), whereas treatment effects increased wasting prevalence by 10% (aPR = 1.1; p = 0.013).

### Conclusion

Wasting is a prevalent form of malnutrition among pediatric cancer patients requiring the integration of nutritional services to address the nutritional needs of children,

**Data availability statement:** All relevant data are within the paper and its Supporting information files.

**Funding:** The author(s) received no specific funding for this work.

**Competing interests:** The authors have declared that no competing interests exist.

especially those aged greater than 5 years, those with cancers along the gastro-intestinal tract and those experiencing treatment effects. Additionally, health and nutrition education programs tailored to the caregiver's level of education are needed.

## Introduction

Pediatric cancer, a primary contributor to childhood mortality globally assumes increased significance in low-resource settings such as Africa, where over 80% of cases are reported [1,2]. The continent also faces a high burden of malnutrition, which significantly influences the efficacy of cancer treatments, patient quality of life, and survival and is solely responsible for more than 20% of mortalities [2–4]. For example, in South Africa, 31.7% of children with cancer were malnourished, 17.3% were wasted, 7.2% were stunted, and another 7.2% were both wasted and stunted [5]. Similarly, a retrospective study in Uganda revealed that 34.6% of children with cancer had acute malnutrition at diagnosis, with 54.3% classified as moderate and 45.7% as severe, highlighting the rapid onset of wasting in this group [6].

Wasting prevalence is aggravated by several factors, including the type and location of cancer, its stage, the treatment regimen, dietary intake, treatment side effects, and the child's age [7–9], as well as health system factors [6]. Uganda currently reports approximately 3000 pediatric cancer cases annually, with only 30% of these patients receiving appropriate treatment at cancer centres [10]. Children undergoing treatment and those diagnosed with the disease face numerous challenges, including wasting [2,5,11]. Children who experience wasting endure a reduced quality of life, treatment intolerance, non-adherence, non-responsiveness, relapse, and increased mortality [12–15]. While some studies in Uganda have attempted to assess the prevalence of wasting and its associated factors, these efforts have faced limitations. One study, for example, used a sample size of only five patients to conclude a 60% malnutrition prevalence [1]; this sample size was not representative of the study population. Another study [6] relied exclusively on weight-based tools, which can falsely categorize children with tumors as having normal weight or even obesity. To address the limitations associated with weight-based anthropometric assessments in pediatric cancer patients—particularly those with solid tumors, which may distort body weight and lead to misclassification of nutritional status—this study utilized Mid-Upper Arm Circumference (MUAC) as the primary tool for assessing wasting. MUAC is a non-weight-dependent anthropometric measure that assesses muscle mass and fat stores, making it a more reliable indicator of acute malnutrition in settings where body weight may be affected by tumor burden or fluid retention. Studies have demonstrated that MUAC is more accurate and predictive of mortality risk than weight-for-age or BMI-for-age in children with clinical complications, including cancer [16]. Furthermore, a 2022 study by Ssebbiri at the Uganda Cancer Institute focused on children aged 5–14 years and assessed only dietary intake, yet the average age range for pediatric cancer patients typically lies between 3 and 7 years [17,18], potentially excluding a significant portion of the affected population. Therefore, to

directly respond to the gaps identified in earlier studies, the current study determined the prevalence of wasting and its associated factors among pediatric cancer patients aged 2–17 years at the Uganda Cancer Institute with the use of MUAC as the tool of assessing wasting. Our study applied MUAC cutoffs aligned with WHO-recommended z-score categories— ≥ −1 (no wasting), −1 to −1.9 (risk of wasting), −2 to −2.9 (moderate wasting), and ≤ −3 (severe wasting)—to provide a more accurate and context-appropriate assessment of nutritional status [16].

## Materials and methods

### Ethical Considerations

The study was conducted following approval from Makerere University, Higher Degrees, Research and Ethics Committee (Protocol number MakSPH-REC-382) and permission provided by the Uganda Cancer Institute (UCI) SR-16/24. The parents/ guardians were given a consent form which they signed as an indicator of willingness to participate in the study. The collected information was treated with the utmost confidentiality before and during the analysis process.

### Study area

The study was conducted at the Uganda Cancer Institute (UCI) 5 kilometres from Uganda's capital city, Kampala. The UCI was established in 1967 to deliver cancer treatment, research, and training to healthcare professionals. Consequently, the Institute also serves as a tertiary, specialized, teaching and research centre affiliated with Makerere University School of Medicine and Mulago National Referral Hospital. Currently, UCI serves as the East African centre of excellence in oncology as well as the national hospital for cancer and offers comprehensive treatment and care for cancer patients from all corners of the country and across East Africa. The pediatric oncology department offers specialized cancer prevention and management services to children, including cancer screening, early diagnosis, staging, chemotherapy, radiation therapy, and surgery. The Pediatric Unit has a team of qualified pediatric oncologists, nurses, a nutritionist, and other healthcare professionals who collaborate to serve approximately 40 inpatients and, on average, 150 patients daily (in- and outpatients).

### Study design and population

The study utilized a cross-sectional design and adhered to the Strengthening the Reporting of Observational Studies in Epidemiology (STROBE) guidelines [19]. The study population consisted of caregiver–child pairs (caregivers of children aged 2–17 years) at both the outpatient unit and the inpatient unit of the Uganda Cancer Institute. The study was also guided by the conceptual framework adopted from the UNICEF conceptual framework, which explains the factors that influence the different forms of malnutrition [17]. The factors highlighted in the framework include enabling factors; social cultural norms; hospital resources; underlying factors; disease effects and treatment effects; and intermediate factors such as disease and diet. The use of this framework provides an understanding of the factors associated with wasting.

### Sample size determination and sampling

The sample size of 384 was calculated using the modified Kish–Leslie formula (1965) [18] for a single population, with a 34.6% proportion of acute malnutrition among pediatric patients at the Mbarara Regional Referral Hospital Institute [6] and an error of 5% (1.96) and 95% confidence level. The study employed a systematic random sampling technique with a sampling interval of 3. The sampling interval was calculated on the basis of the number of patients (900) expected by health workers in 4 months divided by the calculated sample size (384) ($k = 900/384 = 2.34 = 3$). The random start was 1, and the other participants were selected after every 3 individuals [20]. Individuals who did not consent or those who were re-attendants were skipped and replaced by the next eligible participant. A sample size of 270 child–caregiver pairs approximately 4 months between May and August 2024.

## Data collection

Data were collected through face–to-face interviews using a structured questionnaire. Caregiver–child pairs were asked about social demographics; here, they provided information on their age, sex, caregiver's marital status and employment status. Age was recorded in complete years for both the children and the caregivers. They also provided information on the child's dietary intake through both 24-hour recalls of 11 food groups and seven-hour recalls (7-day food frequency questionnaire (FFQ). While conducting the FFQ, the dietary intake of 59 foods was assessed because during the pre-test, it was found that most of these foods were consumed. Furthermore, the number of days a food was eaten, such as 2 days and 3 days, was recorded. With the use of mid-upper arm circumference tapes (for children younger than 5 years and those above), the anthropometric data for each child were taken and recorded twice as measurements to obtain an average. These MUAC measurements were taken on the basis of the IIPAN and WHO guidelines [16]. Furthermore, information on cancer-related factors, such as the type and stage of cancer, was retrieved from the patients' hospital records to ensure accuracy and consistency. At this point, only one treatment therapy was considered for children who were on more than one treatment therapy such as radiotherapy and chemotherapy. The stages of cancer were recorded as stage I, stage II, stage III, stage IV, and unstaged. Information on treatment effects was mostly self-reported. Patient– caregiver pairs were asked about the treatment effects the child experienced. However, while the diagnosis was reported, any treatment effect that was further reported in the records that was not self-reported was included. These treatment effects included vomiting, diarrhea, gastrointestinal obstruction, loss of appetite, and taste aversion. To measure the outcome variable wasting, MUAC readings were taken. The measurement was done using MUAC tapes which were colour coded, flexible, non-tearable and non-stretchable. Tapes of measuring ranges of 26.5 cm, 40.5 cm, and 45.5 cm were used for children 6–59 months and 6–10 years, adolescents 10–15 years and 15–18 years respectively. The MUAC tapes were standard (S0145620 MUAC, Child 11.5 Red/ PAC- 50) supplied by UNICEF. During MUAC measurement, the less active arm was identified and then flexed at a 90° angle. The elbow and the acromion of the shoulder were determined. With the MUAC tape, the spot where the elbow and shoulder joints meet in the middle was identified. The tape was then placed around the upper arm midway while the arm was relaxed and the torso was hanging down. For quality control, the tension of the tape was properly assessed to ensure that it was neither too loose nor too tight to allow accurate readings (the nearest 0.1 cm). Considering the cut-off points from the World Health Organization child growth cut-off points and Mramba's study (which is aligned with the WHO MUAC for age Z scores), the average result obtained was later categorized into no wasting (no wasting, at risk of wasting) and wasting (severe wasting and moderate wasting, at risk) [21,22] .

## Data management and analysis

After data collection, the questionnaires were checked for completeness and accuracy before leaving the field, reviewed during data entry and cleaned to ensure consistency and completeness. For categorical variables, appropriate encoding schemes (yes was represented by 1, whereas no was represented by 0) were used to ensure numerical representation while maintaining the integrity of the underlying information. The collected data were entered into STATA version 14.0 for Windows for analysis. Some of the 59 foods were then grouped into nine food groups as per the study by Steyn and colleagues [23]. The categorization was performed following World Food Program analysis tutorials for dietary diversity scores in which different foods were combined into 3 food groups [16]. This approach was adopted from the World Food Programme [16]. However, some foods which were not under these 9 food groups were left out; this is what was considered; Starchy; bread or macron + rice/rice products + sweet potatoes + irish potatoes + cassava + cassava flour + matooke + yams + maize + posho + sorghum + millet, diary; milk + diary. Organ meat; kidney + offals + liver, eggs; eggs, flesh foods; poultry + fish + meat), legumes; (beans + peas + groundnuts + nuts), vitamin A vegetables; sukumawiki + cabbage + nakatti + amaranthus + pumpkin leaves + pumpkin), vitamin A fruits; pawpaw + mangoes + bananas + jackfruit), other fruits vegetables; tomatoes + oranges + passion fruits + pineapples + avocado + watermelon + apples + grapes + cabbage. The

responses, that is, the days of consumption, were also grouped into 3 categories: 1 = 0 days, 2 = 1–6 days and 3 = 7 days or more, according to the World Food Program [16]. The children's age was categorized into less than 5 years and greater than 5 years to reflect the common age groupings in nutrition, as guided by a study done in Malawi to determine the burden of malnutrition in childhood cancer [24]. The age of the caregivers was grouped according to the mean age.

Furthermore, the normally distributed continuous variables were summarized using means and standard deviations, and the categorical variables were summarized using frequencies and percentages. The dependent variable was summarized as a proportion with a 95% confidence interval. For regression analysis, the dependent variable was dichotomized as 1 = yes and 0 = no. Since the prevalence of wasting was greater than 10%, modified Poisson regression was used to determine the associated factors with prevalence ratios as the measure of association. For bivariate analysis, all variables with p values < 0.25 were considered for multivariable analysis. However, the correlations between variables were assessed by multi-collinearity analysis, and variables with a ratio of ≥ 0.4 were not considered for multivariable analysis. During multivariable analysis, the manual backward elimination method was used. Statistical significance was declared at a p-value ≤ 0.05.

## Results

### Sociodemographic characteristics of caregiver–child pairs of pediatric cancer patients

Table 1 summarizes the sociodemographic characteristics of the study participants. A total of 270 caregivers–child pairs who were available during the study period were included in this study. The mean (SD) age of the children was 9.6 (±4.2) years, and the majority (55.9%) were male. For the caregivers, the mean (SD) age was 35.4 (±0.7) years, and the majority (64.1%) were married. Primary education was the highest level of education attained by most (41.9%) caregivers, and 81.5% were employed.

### Clinical characteristics of pediatric cancer patients

Table 2 presents the clinical characteristics of the pediatric cancer patients. Most of the pediatric cancer patients (73.3%) had solid cancers. Wilms tumor (13.3%) was the most common solid cancer, whereas acute lymphoblastic leukemia was the most common (12.6%) type of liquid cancer. The types of cancer were mainly located away from the GIT (73.3%), not staged (76.0%) and treated by chemotherapy (87.0%).

### Treatment effects experienced by pediatric cancer patients aged 2–17 years at the UCI

Over 80% of the children who experienced treatment effects had a reduced appetite, and only 48.9% and 34.1% experienced vomiting and mucositis, respectively. Fewer than 30% of the participants experienced diarrhea, taste aversions, constipation, and obstruction. This is highlighted in Fig 1.

### Dietary intake of pediatric cancer patients aged 2–17 years at the Uganda Cancer Institute

Over 90% of pediatric cancer patients consumed 4 or more food groups in the past 24 hours.

Analysis of the food consumption of pediatric cancer patients over 7 days revealed that more than half consistently consumed three food groups daily. Starchy foods were the most commonly consumed (96.3%), followed by fruits and vegetables (77.4%) and legumes (52.5%). Four additional food groups were consumed for 1–6 days: flesh meat (67.0%), eggs (62.2%), vitamin A-rich fruits (58.2%), and dairy (51.5%).

### Prevalence of wasting among pediatric cancer patients aged 2–17 years at the UCI

Among the 270 pediatric cancer patients assessed, 27.4% (n = 74) were wasted. Approximately half of the children were not wasted, 18.1% were at risk of wasting, and 15.9% were severely wasted.

**Table 1. Sociodemographic characteristics of 270 pediatric cancer patients and their caregivers.**

| Variable | Frequency (n) | Percentage (%) |
|---|---|---|
| Category | N = 270 | |
| **Child's sex** | | |
| Male | 151 | 55.9 |
| Female | 119 | 44.1 |
| **Child's age** | Mean (SD)=9.60(±4.20) | |
| below 5 years | 59 | 21.9 |
| 5 years and more | 211 | 78.25 |
| **Caregiver's sex** | | |
| Male | 85 | 31.5 |
| Female | 185 | 68.5 |
| **Child's ward** | | |
| Outpatient ward | 230 | 85.2 |
| Inpatient ward | 40 | 14.8 |
| **Caregiver's age** | Mean (SD)=35.4(±0.7) | |
| Below 35 years | 136 | 50.4 |
| 35 years and above | 134 | 49.6 |
| **Caregiver's level of education** | | |
| Primary | 113 | 41.9 |
| Secondary | 102 | 37.8 |
| Tertiary | 42 | 15.6 |
| None | 13 | 4.8 |
| **Caregiver's marital status** | | |
| Married | 173 | 64.7 |
| Unmarried | 97 | 35.9 |
| **Physiological status of the caregiver** | | |
| Pregnant | 10 | 3.7 |
| Breast-feeding | 16 | 5.9 |
| Pregnant and Breastfeeding feeding | 2 | 0.7 |
| Not pregnant/breastfeeding | 157 | 58.2 |
| Not applicable | 85 | 31.5 |
| **Caregiver's occupation status** | | |
| Employed | 220 | 81.5 |
| Unemployed | 50 | 18.5 |

### Factors associated with wasting among pediatric cancer patients aged 2–17 years at UCI

At bivariate analysis, there was a statistically significant association between the following variables: age of the child, treatment effects, caregivers' level of education, caregiver's marital status and wasting. These variables and any other variables that had a p-value < 0.25 were considered for multivariable analysis. After adjusting for confounding factors in the multivariable analysis, the age of the child (children 5 years and older), cancer location, caregiver's level of education, and treatment effects emerged as statistically significant predictors of wasting. Children aged 5 years and above had a 20% higher prevalence of wasting compared to those under 5 years of age (aPR = 1.2; 95% CI: 1.1–1.3; p = 0.008). Children whose caregivers attained tertiary education predicted a 20% lower prevalence of wasting (aPR = 0.8; 95% CI: 0.8–0.9; p = 0.002) compared to those whose caregivers had completed only a primary level of education. Similarly, children with

**Table 2. Clinical characteristics of pediatric cancer patients aged 2-17 years at the UCI.**

| Variable | Frequency (n) | Percentage (%) |
|---|---|---|
| Categories | **N = 270** | |
| **Child's ward** | | |
| Outpatient ward | 230 | 85.2 |
| Inpatient ward | 40 | 14.8 |
| **Type of cancer** | | |
| Solid tumor | 198 | 73.3 |
| Liquid tumor | 72 | 26.7 |
| **Type of solid or liquid cancer** | | |
| Acute Lymphoblastic leukemia | 34 | 12.6 |
| Acute myeloid Leukemia | 23 | 8.5 |
| Wilms Tumor | 36 | 13.3 |
| Hodgkins Lymphoma | 18 | 6.7 |
| Non-Hodgkin Lymphoma | 10 | 3.7 |
| Rhabdomyosarcoma | 26 | 9.6 |
| Ewing Sarcoma | 5 | 1.9 |
| Osteosarcoma | 22 | 8.2 |
| Retinoblastoma | 12 | 4.4 |
| Burkitt's Lymphoma | 7 | 2.6 |
| Others | 32 | 11.9 |
| Kaposi Sarcoma | 11 | 4.1 |
| Brain Tumor | 10 | 3.7 |
| Nasal Pharyngeal Carcinoma | 8 | 3.0 |
| Neuroblastoma | 4 | 1.5 |
| Rosai Dofman Disease | 4 | 1.5 |
| Small round blue cell tumor | 4 | 1.5 |
| Synovial Sarcoma | 4 | 1.5 |
| **Cancer location** | | |
| Along/near the GIT | 72 | 26.7 |
| away from the GIT | 198 | 73.3 |
| **Cancer stage** | | |
| Stage 1 | 8 | 3.0 |
| Stage 2 | 7 | 2.6 |
| Stage 3 | 21 | 7.8 |
| Stage 4 | 37 | 13.7 |
| Unstaged | 197 | 73.0 |
| **Treatment type** | | |
| Chemotherapy | 235 | 87.0 |
| Radiotherapy | 12 | 4.4 |
| Surgery | 23 | 8.5 |
| **Treatment effects** | | |
| No | 44 | 16.3 |
| Yes | 226 | 83.7 |

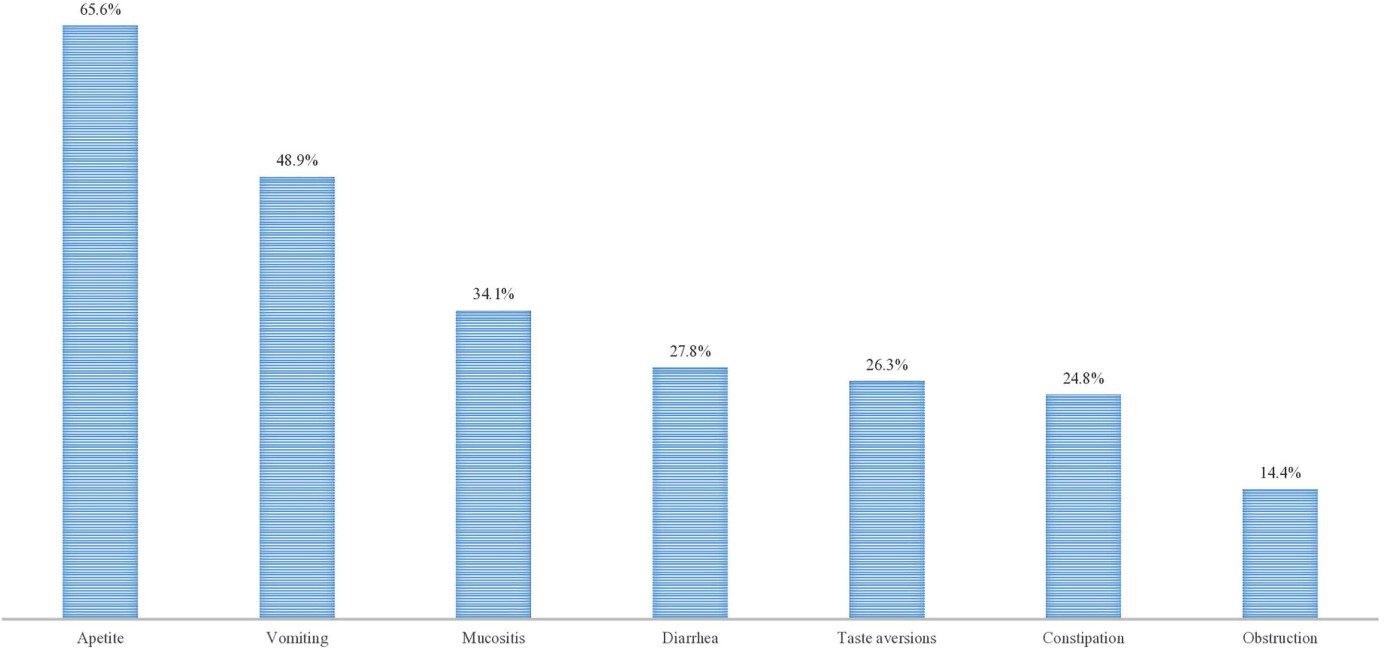

**Fig 1. Treatment effects experienced by pediatric cancer patients aged 2--17 years at UCI.**

cancers located along or near the gastrointestinal tract had a 10% greater prevalence of wasting (aPR = 1.1; 95% CI: 1.0–1.2; p = 0.028) than those with cancers located away from the gastrointestinal tract. Additionally, a 10% greater prevalence of wasting was observed among children who experienced treatment side effects (aPR = 1.1; 95% CI: 1.0–1.3; p = 0.013) than among those who did not experience any treatment effects (Table 3).

## Discussion

This cross-sectional study aimed to determine the prevalence of wasting and its associated factors among pediatric cancer patients aged 2–17 years at the Uganda Cancer Institute in Kampala, Uganda.

The study revealed a 27.4% prevalence of wasting among pediatric cancer patients. The factors statistically associated with wasting were being >5 years of age, having a caregiver with a tertiary education level, having a tumor located along the gastrointestinal tract, and experiencing treatment effects.

The prevalence of wasting among pediatric cancer patients was 27.4%, indicating a significant proportion of affected patients. This prevalence is lower than the 60% reported by Abdoulie, who assessed five pediatric cancer patients at the UCI [1], and lower than the findings of Jeanine and colleagues, who retrospectively evaluated acute malnutrition among children under 15 years of age at Mbarara Regional Referral Hospital in Uganda [6]. The differences could be because the current study was conducted at a national referral hospital where the most severe cases are likely to be found, which may be few hence resulting into a lower wasting prevalence in this study. Secondly, the current study used the MUAC for age Z scores, a more specific tool for nutritional status assessment that could more specifically identify wasting children [25,26]. In contrast, the current prevalence of wasting is higher than the 17.3% reported by Geddara in a study conducted among pediatric cancer patients in South Africa [5]. This disparity could be attributed to South Africa's more advanced health-care infrastructure and better nutritional support services than Uganda does [5,27]. This difference highlights the need for improved healthcare services with a multidisciplinary approach to integrate comprehensive nutrition into standard cancer care to reduce wasting.

**Table 3. Factors associated with wasting among pediatric cancer patients aged 2-17 years at the Uganda Cancer Institute.**

| Variable | No wasting | Wasting | unadjusted Prevalence Ratio (uPR) (95%CI) | P-Value | adjusted Prevalence Ratio (aPR) (95%CI) | P-Value |
|---|---|---|---|---|---|---|
| **Category** | | | | | | |
| **Sex of the child** | | | | | | |
| Male | 105(69.5) | 46(30.5) | 1 | | 1 | |
| Female | 91(76.5) | 28(23.5) | 1.0(0.9-1.0) | 0.200 | 1.0(1.0-1.0) | 0.291 |
| **Child's age** | | | | | | |
| below 5 years | 50(84.8) | 9(15.3) | 1 | | 1 | |
| 5 years and more | 146(69.2) | 65(30.8) | 1.1(1.0-1.3) | **0.008*** | 1.2(1.1-1.3) | **0.002*** |
| **Child's ward** | | | | | | |
| Inpatient ward | 21(7.8) | 19(7.0) | 1 | | | |
| Outpatient ward | 175(64.8) | 55(20.4) | 1.2(0.1-0.4) | **0.223** | | |
| **Caregivers' sex** | | | | | | |
| Male | 62(72.9) | 23(27.1) | 1.0(0.9-1.0) | 0.930 | | |
| Female | 134(72.4) | 51(27.6) | 1 | | | |
| **Caregivers' age** | | | | | | |
| 34 and below | 102(75) | 34(25) | 1.0(1.0-1.1) | 0.370 | | |
| 35 and above | 94(70.2) | 40(29.9) | 1 | | | |
| **Caregivers' level of education** | | | | | | |
| Primary | 75(66.4) | 38(33.6) | 1 | | 1 | |
| Secondary | 75(73.5) | 27(26.5) | 1.0.(1.0-1.0) | 0.252 | 1.0(0.9-1.1) | 0.362 |
| Tertiary | 36(85.7) | 6(14.3) | 0.9(0.81.0) | **0.007*** | 0.8(0.8-0.9) | **0.002*** |
| None | 10(76.9) | 3(23.1) | 0.9(0.8-1.1) | 0.414 | 1.0(0.8-1.2) | 0.644 |
| **Caregivers' marital status** | | | | | | |
| Married | 133(76.9) | 40(23.1) | 1 | | | |
| Unmarried | 63(65.0) | 34(35.1) | 1.2(1.0-1.2) | **0.040*** | | |
| **Caregivers' occupation status** | | | | | | |
| Employed | 158(71.8) | 62(28.2) | 1.0(1.0.-1.1) | 0.540 | | |
| Unemployed | 38(76) | 12(24) | 1 | | | |
| **Type of cancer** | | | | | | |
| Solid tumor | 142(71.7) | 56(28.3) | 1.0(0.9-1.1) | 0.540 | | |
| Liquid tumor | 54(75) | 18(25) | 1 | | | |
| **Cancer location** | | | | | | |
| Along/near the GIT | 48(66.7) | 24(33.3) | 0.9(0.9-1.0) | 0.197 | 1.1(1.0-1.2) | **0.028*** |
| away from the GIT | 148(74.8) | 50(25.3) | 1 | | 1 | |
| **Cancer stage** | | | | | | |
| stage 1 | 5(62.5) | 3(37.5) | 1 | | | |
| stage 2 | 5(71.4) | 2(28.6) | 1.0(0.8-1.5) | 0.713 | | |
| stage 3 | 17(81.0) | 4(19.1) | 0.9(0.7-1.3) | 0.611 | | |
| stage 4 | 25(67.6) | 12(32.4) | 1.0(0.8-1.4) | 0.839 | | |
| Unstaged | 144(73.1) | 53(26.9) | 1.0(0.8-1.3) | 0.923 | | |
| **Treatment type** | | | | | | |
| Chemotherapy | 171(72.8) | 64(27.2) | 1 | | | |
| Radiotherapy | 9(75) | 3(25) | 0.1(0.8-1.2) | 0.860 | | |
| Surgery | 16(69.6) | 7(30.4) | 1.0(0.9-1.2) | 0.750 | | |
| **Treatment effects** | | | | | | |
| No | 38(86.4) | 6(13.6) | 1 | | 1 | |

*(Continued)*

**Table 3.** (Continued)

| Variable | No wasting | Wasting | unadjusted Prevalence Ratio (uPR) (95%CI) | P-Value | adjusted Prevalence Ratio (aPR) (95%CI) | P-Value |
|---|---|---|---|---|---|---|
| Yes | 158(70.0) | 68(30.1) | 1.1(1.0-1.3) | **0.008*** | 1.1(1.0-1.3) | **0.013*** |
| **Individual Dietary Score** | | | | | | |
| <4 food groups | 11(57.9) | 8(42.1) | 1 | | | |
| 4 and more food groups | 185(73.7) | 66(26.3) | 1.1(1.0-1.3) | 0.160 | | |

The prevalence of wasting was higher among children aged 5 years and above than among those under the age of 5 years, suggesting that older pediatric cancer patients are at greater risk of wasting. Similar trends were observed in a study conducted at Mbarara Regional Referral Hospital [6]. Other studies in Malawi [24], Ghana [25] and Nigeria [27] also reported higher wasting prevalence in children over 5 years. In low- and middle-income countries, most nutritional programs focus primarily on children aged less than 5 years [24,25,27], which could explain this finding. Therefore, it is crucial to provide tailored nutritional support for pediatric cancer patients of all ages to minimize wasting.

Children with cancers located along or near the gastrointestinal tract (GIT) presented a higher wasting prevalence than did those with cancers located away from the GIT. This may be attributed to the direct impact of tumors on food intake, digestion, nutrient absorption, and overall gastrointestinal function because they can physically occupy space [15,2]. This finding is supported by studies conducted in Athens and Brazil, which also reported that patients with GIT tumors were more likely to be malnourished [28,29]. This, therefore, highlights the need to implement and adhere to tailored nutritional interventions such as parenteral nutrition, as recommended by the International Initiative for Pediatric and Nutrition (IIPAN) [30] as well as considering immunological nutritional care.

The study findings indicated that the prevalence of wasting was lower among children whose caregivers had attained a tertiary level of education than among those whose caregivers had only a primary education. In a study in Ethiopia, a caregiver's (particularly the mother's) education level was associated with more wasting among children [31]. Therefore, there is a need to assess how nutritional education programs are conducted at UCI among caregivers to understand how best they can be conducted to lower the prevalence of wasting among children whose caregivers are at the primary level of education.

The study revealed that children who experienced treatment effects had a higher prevalence of wasting than did those who did not. This could be because the effects experienced by children, such as reduced appetite, vomiting mucositis, diarrhea, taste aversions, constipation, or gastrointestinal obstruction, all hamper food intake and food absorption. These findings align with those of previous studies, which highlight how chemotherapy-related side effects, such as vomiting and mucositis, reduce food intake and contribute to wasting [7]. This could be because effects such as mucositis hinder nutrient intake by causing painful mouth sores [7,32]. According to Sumdarmanto & Primativa, diarrhea, although less reported in this study, disrupts nutrient absorption [15,2]. A reduced appetite, reported by more than half of the children, is a critical factor leading to decreased food intake, similar to the findings of previous studies from Tanzania and Kenya [8]. Therefore, integrating comprehensive nutritional support into routine cancer treatment is pivotal in managing treatment related side effects and reducing wasting among pediatric cancer patients. Strategies to manage side effects may include; 1) encouraging adequate rehydration with fluids to manage diarrhea, 2) maintaining small but frequent nutrient dense meals to improve appetite, 3) administering parenteral nutrition for patients with gastrointestinal obstruction, 4) providing nutrition education and counselling, and 5) promoting consumption of a high fibre diet to relieve constipation.

## Study strengths and limitations

A key strength of this study was the use of a non-weight-based nutritional assessment tool (MUAC for age Z scores), which increased the accuracy of identifying wasted children [22,25]. However, this study has several limitations, such as the inclusion of a sample size of 270 which may restrict representativeness. Failure to achieve the desired sample size is attributed to exclusion of participants with incomplete/ missing records as well as the limited number of patients that seek care at UCI. Caregiver recall bias should also be expected given the cross-sectional design of the study. The limitation to a single institution and the cross-sectional study design may limit the generalizability of results and inference of causality.

## Conclusion

Wasting is a common yet modifiable malnutrition condition among pediatric cancer patients at the Uganda Cancer Institute, where its prevalence is 27.4%. The key factors associated with wasting were the age of the child, caregiver's level of education, treatment effects, and cancer location. Therefore, there is a need to further focus on the needs, especially the nutritional needs, of children aged greater than 5 years, those who have tumors located along the gastrointestinal tract, and those experiencing treatment effects.

## Supporting information

**S1 Data. Raw Data.**
(XLSX)

## Acknowledgments

I would like to acknowledge all the study participants who contributed data to this study amidst their difficult circumstances.

## Author contributions

**Conceptualization:** Daisy Wannyana, Arthur Bagonza, Sandrah Joyce Mwima, Rawlance Ndejjo.

**Data curation:** Daisy Wannyana, Arthur Bagonza, Sandrah Joyce Mwima, Christine Nalwadda, Rawlance Ndejjo.

**Formal analysis:** Daisy Wannyana, Arthur Bagonza, Sandrah Joyce Mwima, Rawlance Ndejjo.

**Investigation:** Daisy Wannyana, Arthur Bagonza, Christine Nalwadda.

**Methodology:** Daisy Wannyana, Arthur Bagonza, Christine Nalwadda, Rawlance Ndejjo.

**Project administration:** Daisy Wannyana.

**Resources:** Daisy Wannyana.

**Software:** Christine Nalwadda.

**Supervision:** Sandrah Joyce Mwima, Rawlance Ndejjo.

**Validation:** Sandrah Joyce Mwima, Christine Nalwadda, Rawlance Ndejjo.

**Writing – original draft:** Daisy Wannyana, Christine Nalwadda, Rawlance Ndejjo.

**Writing – review & editing:** Daisy Wannyana, Arthur Bagonza, Sandrah Joyce Mwima, Christine Nalwadda, Rawlance Ndejjo.

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
