## [Decision Letter · Decision Letter 0]

21 Jun 2025

Dear Ms. Wannyana,

Thank you for submitting your manuscript to PLOS ONE. After careful consideration, we feel that it has merit but does not fully meet PLOS ONE’s publication criteria as it currently stands. Therefore, we invite you to submit a revised version of the manuscript that addresses the points raised during the review process.

We look forward to receiving your revised manuscript.

Kind regards,

Deogratias Munube

Academic Editor

PLOS ONE

Journal Requirements:

4. Please ensure that you refer to Figure 1 in your text as, if accepted, production will need this reference to link the reader to the figure.

5. Please upload a copy of Figure 2, to which you refer in your text on page 14. If the figure is no longer to be included as part of the submission please remove all reference to it within the text.

6. Please remove all personal information, ensure that the data shared are in accordance with participant consent, and re-upload a fully anonymized data set.

Reviewers' comments:

Reviewer's Responses to Questions

**Comments to the Author**

1. Is the manuscript technically sound, and do the data support the conclusions?

Reviewer #1: Yes

Reviewer #2: Yes

2. Has the statistical analysis been performed appropriately and rigorously?

Reviewer #1: Yes

Reviewer #2: Yes

3. Have the authors made all data underlying the findings in their manuscript fully available?

Reviewer #1: Yes

Reviewer #2: Yes

4. Is the manuscript presented in an intelligible fashion and written in standard English?

Reviewer #1: Yes

Reviewer #2: Yes

Reviewer #1: 1. Research Methodology:

Is the cross-sectional approach used in this study sufficient to establish causal relationships between factors influencing wasting in pediatric cancer patients, or would a longitudinal study design provide a better understanding of these relationships?

Is the sample size of 270 patients large enough to be representative of the overall pediatric cancer population in Uganda?

2. Variables Used:

The study mentions treatment side effects as a factor influencing wasting. However, were other factors such as initial nutritional status, cancer severity, or type of cancer therapy considered as important variables in the analysis?

Can you provide more detailed information on the types and stages of cancer present in the study sample? Do these factors directly correlate with wasting prevalence?

3. Social and Economic Aspects:

The study identifies parental education level as a factor affecting wasting prevalence. How about other socio-economic factors, such as family economic status or access to healthcare? Were these factors included in the analysis?

4. Nutritional Data Collection:

What methods were used to collect data on dietary intake in this study? Are the interviews or questionnaires used to gather dietary information from parents or caregivers accurate enough to reflect the child's true diet?

Is there a possibility of bias in the reported dietary intake, considering it relies on parental or caregiver reports?

5. Clinical Recommendations:

Based on the findings, are there more specific recommendations for interventions to prevent or manage wasting in pediatric cancer patients, especially concerning the management of treatment side effects?

Does this study provide recommendations for public health policies or improvements in clinical practices that can be implemented at cancer centers or hospitals in Uganda?

6. Demographic Differences:

The study does not mention whether there are significant differences in wasting prevalence based on other demographic factors, such as gender or ethnicity of the patients. How do these differences impact wasting prevalence?

7. Application of Study Results:

Can the results of this study be widely applied to pediatric cancer patients in other developing countries, or are there local factors that need to be considered before generalizing the study's findings?

Reviewer #2: Reviewer: Nicolette Nabukeera Barungi

Comments

This is a good study and very relevant. However, there should be revisions as below:

The authors need to be clear about reference 6 which was a study done in Uganda with higher numbers, the authors need to state the setting and time when that study was done so as to justify why they did a similar study. What gaps did they find in this study?

Line 93-96, Authors need to be more clear on the gaps of using weight to determine nutritional status. They should phrase the objective better, indicating that in order to overcome the errors introduced by using weight, they determined nutritional status. Before this, they need a line or 2 on what MUAC assesses with references, its accuracy compared to Weight for age etc.

MUAC has many issues when used for children aged 5 years and above. It is quite unreliable and cut offs were developed in refugee settings and its reliability is also problematic. The authors need to discuss this. It would be good to categorize the ages, below 5 and above 5 years. Almost 80% were above 5 years in your study so this is very concerning. How many had solid tumors?

We need to know the numbers of children at UCI like monthly or annually. It is more informative than the daily attendances and inpatients at any one point. How many inpatients per yearor per month?

Does the unit have general doctors and general paediatricians? Role of nutritionist? Is there routine assessment and education or supplementation? Focus the study setting to what affects the study results and helping with generalizability.

Why child-caregiver pairs and not just the child? What if the child had multiple care takers?

Line 129 has ref 10 yet earlier that study was ref 6. Are you using manual referencing? The two are the same reference.

Line 134, “individuals who did not assent”? How about those who did not consent?

Line 148: MUAC was the study outcome but it has not been described in the methods. Which MUAC tapes were used for who? (Seen part of it in line 160-165). It needs to be in one place, not scattered in different places.

IIPAN is not described in full in the manuscript. What is it and why should it be used and not just the WHO?

Line 150: I thought your inclusion criteria was confirmed cancer. If so, then you would not need to exclude suspects because they do not qualify.

Mramba’s study s ref 23 not 22

It is not clear at what point you enrolled the children. How did you handle the admitted children especially asking about the 24 hour recall when they have been on the ward for a while?

Table 1; caregiver employment- does this cover self employment? What does it mean? Father or mother? Or both?

Table 1 or 2 should indicate how many are outpatients and inpatients and they need to be also analyzed as variables affecting nutritional status.

Revise and re-align the references

So many typos, missing full stops etc.

**Do you want your identity to be public for this peer review?** For information about this choice, including consent withdrawal, please see our Privacy Policy

Reviewer #1: No

Reviewer #2: **Yes: ** Nicolette Nabukeera Barungi

---

## [Author Response · Author response to Decision Letter 1]

25 Jul 2025

25th July 2025

Dear Editor,

RE: Response to comments on manuscript titled “FACTORS ASSOCIATED WITH WASTING AMONG PEDIATRIC CANCER PATIENTS AGED 2-17 YEARS AT UGANDA CANCER INSTITUTE: A CROSS-SECTIONAL STUDY”

Thank you for handling the review of our manuscript. We are also grateful to the reviewers who have provided constructive comments that have improved our manuscript. As requested, please see enclosed a point-by-point response for each of the comments provided and a revised version of the manuscript for your further consideration.

Comment

In your method section please include the statement "The parents/ guardians were given a consent form which they signed as an indicator of willingness to participate in the study." in the methods section.

The methods section has been adjusted to include “The parents/ guardians were given a consent form which they signed as an indicator of willingness to participate in the study” (pages 6-7, Lines 107-109).

Academic Editor

Thank you so much for your review.

The manuscript has been aligned to PLOS ONE's style requirements as presented in the links you provided.

Thank you so much for your feedback.

We have improved our methods section to include that we obtained consent from the parents or guardian of the minors; caregivers. The parents/ guardians were given a consent form which they signed as an indicator of willingness to participate in the study. (pages 5-6) (Lines:107-108)

Thank you so much for your feedback

The ORCID iD (0009-0005-4401-5225) of the corresponding author has been validated in the Editorial Manager.

4. Please ensure that you refer to Figure 1 in your text as, if accepted, production will need this reference to link the reader to the figure.

We appreciate your feedback.

Figure 1 in the text has been cross referenced to link the reader to the figure. (Page 15 and Lines: 237-238)

5. Please upload a copy of Figure 2, to which you refer in your text on page 14. If the figure is no longer to be included as part of the submission, please remove all reference to it within the text.

Thank you for the review

The reference “Figure 2” has been removed from the text. (Page 15 and Line: 237)

6. Please remove all personal information, ensure that the data shared are in accordance with participant consent, and re-upload a fully anonymized data set.

Thank you for your comment.

We have not provided raw data for the study. However, the fully anonymized data set has been attached.

Reviewer #1

1. Research Methodology Is the cross-sectional approach used in this study sufficient to establish causal relationships between factors influencing wasting in pediatric cancer patients, or would a longitudinal study design provide a better understanding of these relationships?

We appreciate your feedback.

No, cross sectional studies are not sufficient in establishing causal relationships as indicated by other researchers here. This is because they focus at a single point in time and therefore cannot establish temporality.

In contrast, a longitudinal study design allows observation of changes over time and the establishment of temporal relationships, providing stronger evidence for causal inference. The study’s incomplete data on initial nutritional status measured by Mid-Upper Arm Circumference (MUAC)—a more reliable indicator of wasting in pediatric cancer patients compared to weight-based methods affected by tumors or surgery—limited this potential. Had complete MUAC data been available, a retrospective longitudinal analysis could have offered deeper insights into the progression and determinants of wasting beyond what cross-sectional analysis permit.

Reference

Wang X, Cheng Z. Cross-Sectional Studies: Strengths, Weaknesses, and Recommendations. Chest. 2020 Jul;158(1S):S65-S71. doi: 10.1016/j.chest.2020.03.012. PMID: 32658654.

Is the sample size of 270 patients large enough to be representative of the overall pediatric cancer population in Uganda? Thank you for the review.

Yes, the sample size of 270 is large enough to be representative of the overall pediatric cancer population in Uganda. While, the study enrolled 270 caregiver-child pairs (out of the calculated sample size of 384), the study achieved a 70% response rate which is generally considered acceptable for statistical representativeness. Evidence can be found here.

In this study, the shortfall in sample size is primarily attributed to the exclusion of participants with incomplete or missing treatment records as well as the limited number of pediatric cancer patients that seek treatment at the study area. According to literature (here) only 30% of pediatric cancer patients in Uganda seek care at treatment centers.

Nonetheless, the possibility of sample size being unrepresentative of the overall pediatric cancer population in Uganda has been acknowledged in the study limitations. (page 20, and line 338-342)

References

Ericson A, Bonuck K, Green LA, Conry C, Martin JC, Carney PA. Optimizing Survey Response Rates in Graduate Medical Education Research Studies. Fam Med. 2023 May;55(5):304-310. doi: 10.22454/FamMed.2023.750371. Epub 2023 Feb 21. PMID: 37310674; PMCID: PMC10622096.

https://www.afro.who.int/countries/uganda/news/who-supports-development-child-and-adolescent-cancer-control-strategy-uganda

2. Variables Used: The study mentions treatment side effects as a factor influencing wasting. However, were other factors such as initial nutritional status, cancer severity, or type of cancer therapy considered as important variables in the analysis?

Can you provide more detailed information on the types and stages of cancer present in the study sample? Do these factors directly correlate with wasting prevalence?

Thank you for your feedback.

Yes, some of these factors: stage of cancer, type of cancer, type of cancer therapy were considered as important variables in the analysis. However, no statistically significant association was found between the factors and wasting among pediatric cancer patients at both bivariate and multivariable analysis. Please note that this has been presented in table 3. (Pages 15-17 and lines 270-271)

However, variables that would capture initial nutritional status were not captured. This is due to the incomplete data on initial nutritional status measured by Mid-Upper Arm Circumference (MUAC)—a more reliable indicator of wasting in pediatric cancer patients compared to weight-based methods which are affected by tumors or surgery. (Pages 15-17 and lines 270-271)

The types of cancer were categorized as solid and liquid tumor while the stages were stage I, II, III, IV and Unstaged and included in the analysis.

No, factors; stage of cancer, type of cancer, and type of cancer therapy did not have a statistically significant association with wasting prevalence (table 3). (Pages 15-17 and lines 270-271)

3. Social and Economic Aspects: The study identifies parental education level as a factor affecting wasting prevalence. How about other socio-economic factors, such as family economic status or access to healthcare? Were these factors included in the analysis?

We appreciate your feedback.

Beyond parenteral education level no other socio-economic factors, such as family economic status or access to healthcare were not included in the analysis. Although, we acknowledge the relevance of economic factors, such as economic status, access to health care, to proper nutrition status, we did not collect data on these variables. This is because our study primarily focused on the underlying and immediate factors (treatment-related side effects, cancer type, dietary intake, disease) associated to wasting as per the UNICEF Conceptual Framework on the Determinants of Maternal and Child Nutrition, 2021 which the study adopted.

This therefore, presents the need for future research to adopt a broader approach, incorporating household and contextual variables to better understand the interplay between clinical and socio-economic determinants of wasting among pediatric cancer patients which point we have now highlighted in the manuscript.

Reference

https://www.unicef.org/media/113291/file/UNICEFConceptualFramework.pdf

4. Nutritional Data Collection: What methods were used to collect data on dietary intake in this study? Are the interviews or questionnaires used to gather dietary information from parents or caregivers accurate enough to reflect the child's true diet?

Is there a possibility of bias in the reported dietary intake of pediatric cancer patients, considering it relies on parental or caregiver reports?

We appreciate your valuable feedback.

In this study, dietary intake data for the children were collected using 24-hour recalls covering 10 food groups, along with a 7-day food frequency questionnaire (FFQ), which is one of the standard nutrition assessment tools as it provides the children’s actual diets. Supporting evidence for their validity is available here and here.

Other researchers here and here have also supported the use of caregivers as respondents, especially since the study population included children aged 2 to 17 years, many of whom are not fully capable of independently reporting their dietary intake. Moreover, due to the children’s health conditions, they were often unable or unwilling to respond to the questions themselves, as doing so could be inconvenient or distressing given their experiences.

Nonetheless, the study acknowledges the possibility of recall bias as a limitation. Such bias may result from recall inaccuracies or social desirability effects, where caregivers might unintentionally misreport or modify responses to align with socially acceptable dietary behaviors. (Page 21, Lines: 338-339)

Reference

Wallace A, Kirkpatrick SI, Darlington G, Haines J. Accuracy of Parental Reporting of Preschoolers' Dietary Intake Using an Online Self-Administered 24-h Recall. Nutrients. 2018 Jul 29;10(8):987. doi: 10.3390/nu10080987. PMID: 30060605; PMCID: PMC6115856.

5. Clinical Recommendations:

Based on the findings, are there more specific recommendations for interventions to prevent or manage wasting in pediatric cancer patients, especially concerning the management of treatment side effects?

Does this study provide recommendations for public health policies or improvements in clinical practices that can be implemented at cancer centers or hospitals in Uganda?

Thank you for the review.

Yes, more specific recommendations exist for preventing and managing wasting in pediatric cancer patients, particularly in relation to managing treatment-related side effects. These recommendations emphasize the integration of comprehensive nutritional support into routine cancer care, including strategies such as; 1) encouraging adequate rehydration with fluids to manage diarrhea, 2) maintaining small but frequent nutrient dense meals to improve appetite, 3) administering parenteral nutrition for patients with gastrointestinal obstruction, 4) providing nutrition education and counselling, and 5) promoting consumption of a high fiber diet to relieve constipation. These recommendations have been included in the study. (Page 20 and Lines: 329-335)

Yes, the study provides recommendations for public health policies and clinical practice enhancements that can be applied in cancer centers and hospitals across Uganda. These recommendations include; emphasizing the inclusion of a multidisciplinary collaboration by integrating comprehensive nutritional support into standard cancer care, promoting clinical nutrition, and parenteral nutrition as advocated by the International Initiative for Pediatric and Nutrition (IIPAN). Additionally, the study highlights the need for further research to assess the impact of nutrition interventions and to determine the most effective approaches tailored to the unique requirements of various cancer treatment centers. (Pages 18 and 19, lines 292-294, and 299-302 and 308-311)

6. Demographic Differences: The study does not mention whether there are significant differences in wasting prevalence based on other demographic factors, such as gender or ethnicity of the patients. How do these differences impact wasting prevalence?

Thank you for the review.

No, the current study does not mention whether there are significant differences in wasting prevalence based on other demographic factors, such as gender or ethnicity of the patients. This is because it does not explore differences in wasting prevalence based on ethnicity, as data on this demographic variable were not collected. Although data on gender was collected, the analysis showed no association between gender and wasting prevalence (aPR = 1.0; 95% CI: 1.0–1.0 p-value > 0.291), indicating that gender did not significantly influence the outcome in this study (table 3). Therefore, based on the available data, no meaningful differences in wasting prevalence across these demographic factors were observed. (Pages 15-17 and Lines 270-271)

However other demographic factors including; child’s age, caregivers’ age, sex, level of education, marital status, occupation, physiological status were included in the study. Only child’s age, caregiver’s level of education and marital status showed significant differences in wasting prevalence as illustrated in table 3. (Pages 15-17 and Lines 270-271)

7. Application of Study Results: Can the results of this study be widely applied to pediatric cancer patients in other developing countries, or are there local factors that need to be considered before generalizing the study's findings?

We appreciate your feedback.

Yes, the findings of this study may have broader applicability to pediatric cancer populations in other developing countries, particularly because it was conducted at the Uganda Cancer Institute—a regional center of excellence in cancer care within East Africa. The institute is equipped with advanced diagnostic and treatment facilities and serves as a national referral hospital for cancer care, making its clinical environment and service delivery comparable to other tertiary cancer centers in similar low-resource settings. However, caution is warranted when generalizing the results due to the cross-sectional design of the study, which limits generalizability thus not fully capturing variations in care delivery, nutritional practices, or socio-demographic contexts unique to other regions. (Page 21, and Lines 343-344)

Reviewer 2 The authors need to be clear about reference 6 which was a study done in Uganda with higher numbers, the authors need to state the setting and time when that study was done so as to justify why they did a similar study. What gaps

---

## [Editor Report · Decision Letter 1]

26 Aug 2025

*FACTORS ASSOCIATED WITH WASTING AMONG PEDIATRIC CANCER PATIENTS AGED 2-17 YEARS AT UGANDA CANCER INSTITUTE: A CROSS-SECTIONAL STUDY. (/i>*

Dear Dr. Wannyana,

Thank you for submitting your manuscript to PLOS ONE. After careful consideration, we feel that it has merit but does not fully meet PLOS ONE’s publication criteria as it currently stands. Therefore, we invite you to submit a revised version of the manuscript that addresses the points raised during the review process.

*A rebuttal letter that responds to each point raised by the academic editor and reviewer(s). You should upload this letter as a separate file labeled 'Response to Reviewers'.**A marked-up copy of your manuscript that highlights changes made to the original version. You should upload this as a separate file labeled 'Revised Manuscript with Track Changes'.**An unmarked version of your revised paper without tracked changes. You should upload this as a separate file labeled 'Manuscript'.*

**

We look forward to receiving your revised manuscript.

*Kind regards,*

*Deogratias Munube*

Academic Editor

PLOS ONE

*Journal Requirements:*

*If the reviewer comments include a recommendation to cite specific previously published works, please review and evaluate these publications to determine whether they are relevant and should be cited. There is no requirement to cite these works unless the editor has indicated otherwise. *

Additional Editor Comments:

Dear Author,

Thank you for the extensive responses to the queries. The comment about the sample size calculation and the 70% attainment is a main limitation of the study. Is is possible to request for an amendment to justify the number of children enrolled to the study?

---

## [Author Response · Author response to Decision Letter 2]

1 Sep 2025

26th August, 2025

Dear Editor,

RE: Response to comments on manuscript titled “FACTORS ASSOCIATED WITH WASTING AMONG PEDIATRIC CANCER PATIENTS AGED 2-17 YEARS AT UGANDA CANCER INSTITUTE: A CROSS-SECTIONAL STUDY”

Thank you for handling the review of our manuscript. We are also grateful to the reviewers who have provided constructive comments that have improved our manuscript. As requested, please see enclosed a point-by-point response for each of the comments provided and a revised version of the manuscript for your further consideration.

Comment

In your method section please include the statement "The parents/ guardians were given a consent form which they signed as an indicator of willingness to participate in the study." in the methods section.

The methods section has been adjusted to include “The parents/ guardians were given a consent form which they signed as an indicator of willingness to participate in the study” (pages 6-7, Lines 107-109)

Academic Editor

Current comment from the academic Editor: The comment about the sample size calculation and the 70% attainment is a main limitation of the study. Is it possible to request for an amendment to justify the number of children enrolled to the study?

We thank the reviewer for this important observation. While the calculated sample size was 384 participants, only 270 caregiver-child pairs were enrolled in the study. This is because there was a limited number of eligible pediatric cancer patients attending the Uganda cancer Institute during the four-month data collection period, as well as the exclusion of participants with incomplete or missing records. We have now amended the “methods” and “study strengths and limitations” sections to justify the final sample size achieved and its implications.

Please note that the attached comments were the previous comments from the second reviewer. Please see below;

Reviewer 2 The authors need to be clear about reference 6 which was a study done in Uganda with higher numbers, the authors need to state the setting and time when that study was done so as to justify why they did a similar study. What gaps did they find in this study?

We appreciate your feedback.

Reference 6, conducted by Jeanine et al. (2021), was a retrospective study carried out at the Children’s Cancer Unit of Mbarara Regional Referral Hospital in Uganda. The study analyzed medical records of pediatric cancer patients admitted between May 2017 and 2019, focusing on demographic characteristics, anthropometric measurements at admission, and cancer diagnoses. The study population included children under 16 years of age, and nutritional status was assessed using weight-for-length/height for those under five years, and Body Mass Index-for-age (BMI-for-age) for those aged five years and above.

While this study provided valuable insights, several limitations justify the need for a similar study at the Uganda Cancer Institute (UCI)—the national center of excellence for cancer care. First, the setting of the previous study was a regional referral hospital, whereas UCI is a national referral hospital serving a broader and more diverse population, including more complex and advanced cancer cases. Second, the age range in the study by Jeanine et al. was limited to children below 16 years, therefore excluding older pediatric patients aged 16–17 years, who are also a critical part of the pediatric cancer population. Third, the use of weight-based anthropometric indicators is a limitation, as these measures can be affected by the presence of solid tumors and organomegaly, potentially leading to misclassification of nutritional status.

These methodological and contextual gaps highlight the need for the current study, which aims to provide updated, center-specific data on the prevalence of wasting and its associated factors among pediatric cancer patients aged 2–17 years at UCI, using more appropriate nutritional assessment tools such as the Mid-Upper Arm Circumference (MUAC).

Line 93-96, Authors need to be more clear on the gaps of using weight to determine nutritional status. They should phrase the objective better, indicating that in order to overcome the errors introduced by using weight, they determined nutritional status. Before this, they need a line or 2 on what MUAC assesses with references, its accuracy compared to Weight for age etc.

Thank you for your feedback.

We have clarified on the gaps of using weight to determine nutritional status. (Pages 4-5 and Lines 83-102)

We have rephrased the objective better to indicate that MUAC was used to determine nutrition status in order to overcome the errors introduced by using weight. (Page 5 and lines 96-102)

Information on MUAC assessment, reference values and accuracy compared to weight for age and BMI for age has been added age Sisay et al 2020. (Page 5 and lines 88-102)

Reference

Sisay BG, Haile D, Hassen HY, Gebreyesus SH. Mid-upper arm circumference as a screening tool for identifying adolescents with thinness. Public Health Nutr. 2021 Feb;24(3):457-466. doi: 10.1017/S1368980020003869. Epub 2020 Oct 30. PMID: 33121554; PMCID: PMC10195468.

MUAC has many issues when used for children aged 5 years and above. It is quite unreliable and cut offs were developed in refugee settings and its reliability is also problematic. The authors need to discuss this. It would be good to categorize the ages, below 5 and above 5 years. Almost 80% were above 5 years in your study so this is very concerning. How many had solid tumors?

Thank you for the review.

Several studies including WHO (here, here), Sisay et al 2020 and Mramba et al 2017, indicate that MUAC is comparably as accurate as BMI for age Z-scores among children and adolescents.

Categorization of age into below 5 and above 5 years was done in tables 1 and 3.

(Pages 11-13, 15-17 and Lines 222-223, 270-271 respectively)

According to our study, 73.3 % of the pediatric cancer population had solid tumors as shown in table 2. (Page 13 and Line 231)

Reference

Sisay BG, Haile D, Hassen HY, Gebreyesus SH. Mid-upper arm circumference as a screening tool for identifying adolescents with thinness. Public Health Nutr. 2021 Feb;24(3):457-466. doi: 10.1017/S1368980020003869. Epub 2020 Oct 30. PMID: 33121554; PMCID: PMC10195468.

Mramba L, Ngari M, Mwangome M, Muchai L, Bauni E, Walker AS, Gibb DM, Fegan G, Berkley JA. A growth reference for mid upper arm circumference for age among school age children and adolescents, and validation for mortality: growth curve construction and longitudinal cohort study. BMJ. 2017 Aug 3;358:j3423. doi: 10.1136/bmj.j3423. PMID: 28774873; PMCID: PMC5541507.

We need to know the numbers of children at UCI like monthly or annually. It is more informative than the daily attendances and inpatients at any one point. How many inpatients per year or per month?

Thank you for the review.

Annually, Uganda reports about 3000 new pediatric cancer cases, however, only 30% receive care at cancer centers as shown here. UCI has an inpatient capacity of 40 beds. (Page 4 and Lines 76-78)

Reference

https://www.afro.who.int/countries/uganda/news/who-supports-development-child-and-adolescent-cancer-control-strategy-uganda

Does the unit have general doctors and general pediatricians? Role of nutritionist? Is there routine assessment and education or supplementation? Focus the study setting to what affects the study results and helping with generalizability.

We appreciate your feedback.

Yes, the unit has general doctors, general pediatricians, oncology pediatricians and a nutritionist.

However, the assessment, education and supplementation are not routine due to the high nutritionist: patient ratio. This is because the nutritionist is solely responsible for assessment, education and counselling, supplementation, meal preparation for about 150 children per day hampering establishment of a routine.

In order to align our study to what affects the study results and helping with generalizability, our study setting has been focused to include that currently UCI serves as the East African Centre of excellence in oncology to help with generalizability of the study findings.

Furthermore, the lines 119-122, have captured that the “Pediatric Unit has a team of qualified pediatric oncologists, nurses, a nutritionist, and other healthcare professionals who collaborate to serve approximately, 150 patients daily (in- and outpatients)”. This information highlights the presence of one nutritionist serving 150 pediatric daily which affects the implementation of routine assessment, education and integration of nutrition care into cancer management. (Page 6 and Lines 119-122)

Why child-caregiver pairs and not just the child? What if the child had multiple care takers?

Thank you for your feedback.

Our study considered child caregiver pairs and not just children because parents or caregivers are considered sufficient to accurately to represent their children. Supporting evidence for validity of this choice is available here (Wallace et al 2018).

Furthermore, a proportion of pediatric cancer patients were minors and therefore not fully capable of independently responding to study questions. The physiological status/ wellbeing of some of the pediatric cancer patients also affected their ability to respond to the questions themselves, necessitating the inclusion of caregivers.

Reference

Wallace A, Kirkpatrick SI, Darlington G, Haines J. Accuracy of Parental Reporting of Preschoolers' Dietary Intake Using an Online Self-Administered 24-h Recall. Nutrients. 2018 Jul 29;10(8):987. doi: 10.3390/nu10080987. PMID: 30060605; PMCID: PMC6115856.

Line 129 has ref 10 yet earlier that study was ref 6. Are you using manual referencing? The two are the same reference.

Thank you for the review.

The two references are the same, and the duplicate has been deleted. (Page 23 and Lines 385-387, 399-402)

Line 134, “individuals who did not assent”? How about those who did not consent?

Thank you for the review.

The statement “individuals who did not assent” has been replaced with “individuals who did not consent”. (Page 7 and Line 142)

Line 148: MUAC was the study outcome but it has not been described in the methods. Which MUAC tapes were used for who? (Seen part of it in line 160-165). It needs to be in one place, not scattered in different places.

Thank you for your feedback.

MUAC was the tool of measurement for the outcome variable (wasting). This has been clarified in the method section.

The MUAC tapes used were color coded, flexible, non-tearable and non-stretchable. Tapes of measuring ranges of 26.5cm, 40.5cm, and 45.5cm were used for children 6-59 months and 6-10 years, adolescents 10-15 years and 15-18 years respectively. The MUAC tapes were standard (S0145620 MUAC, Child 11.5 Red/ PAC- 50). (Pages 8-9 and Lines 165-170)

The information on MUAC has been placed in the methods section.

IIPAN is not described in full in the manuscript. What is it and why should it be used and not just the WHO?

Thank you for your feedback.

IIPAN has been described in the manuscript as the International Initiative for Pediatrics and Nutrition.

Headquartered at Columbia University Irving Medical Center in New York City, IIPAN plays a critical role in nutritional training, research, and advocacy for improved quality of life and survival rates of children with cancer.

In Uganda, IIPAN has pioneered the establishment of the nutrition unit at the pediatric cancer ward at the Uganda Cancer Institute.

Line 150: I thought your inclusion criteria was confirmed cancer. If so, then you would not need to exclude suspects because they do not qualify.

Thank you for your feedback.

The exclusion of suspects has been eliminated. (Page 8 and Line 157)

Mramba’s study s ref 23 not 22

We appreciate your feedback.

This has been rectified. Mramba’s study is now ref 20. (Page 25 and Lines 426-429)

It is not clear at what point you enrolled the children. How did you handle the admitted children especially asking about the 24 hour recall when they have been on the ward for a while?

Thank you for your feedback.

Children were enrolled, on basis of being a pediatric cancer patient on either inpatient or outpatient ward.

Children admitted on the ward were still asked about the 24-hour dietary intake despite being on the ward for a while. This is because children are still allowed to eat whether on ward or at home.

Table 1; caregiver employment- does this cover self-employment? What does it mean? Father or mother? Or both?

Thank you for the review.

Yes, caregiver employment does cover self-employment.

A caregiver is any person who is directly taking care of the pediatric cancer patient aged 2-17 years regardless of his/ her relationship with the child. A caregiver may be the child’s father or mother.

Table 1 or 2 should indicate how many are outpatients and inpatients and they need to be also analyzed as variables affecting nutritional status.

Thank you for your feedback.

Table 2 and 3 now indicate how many are outpatients and inpatients were and the analysis of how these variables affect nutritional status has been done. However, no statistically significant association was found between the type of ward the patient was and the prevalence of wasting. (Pages 12-13,15-17 and Lines 231, 270-271 respectively)

Revise and re-align the references

We appreciate your feedback.

This has been rectified

Sincerely,

Daisy Wannyana, on behalf of the co-authors.

---

## [Editor Report · Decision Letter 2]

10 Sep 2025

FACTORS ASSOCIATED WITH WASTING AMONG PEDIATRIC CANCER

PATIENTS AGED 2-17 YEARS AT UGANDA CANCER INSTITUTE: A CROSS-

SECTIONAL STUDY.

PONE-D-25-00802R2

Dear Dr. Wannyana,

We’re pleased to inform you that your manuscript has been judged scientifically suitable for publication and will be formally accepted for publication once it meets all outstanding technical requirements.

Kind regards,

Deogratias Munube

Academic Editor

PLOS ONE

Additional Editor Comments (optional):

Dear Author,

Thank you for addressing all the queries raised by the reviewers.
---

## [Editor Report · Acceptance letter]

PONE-D-25-00802R2

PLOS ONE

Dear Dr. Wannyana,

I'm pleased to inform you that your manuscript has been deemed suitable for publication in PLOS ONE. Congratulations! Your manuscript is now being handed over to our production team.

Kind regards,

on behalf of

Dr. Deogratias Munube

Academic Editor

PLOS ONE